:ᐰ: PLOS | ONE

# Quantitative assessment of interstitial lung disease in Sjögren's syndrome

**Pablo Guisado-Vasco** [ID][1]☉*, **Mario Silva**[2]☉, **Miguel Angel Duarte-Millán**[3]‡, **Gianluca Sambataro**[4]‡, **Chiara Bertolazzi**[5]‡, **Mauro Pavone**[4]‡, **Isabel Martín-Garrido**[1]‡, **Oriol Martín-Segarra**[1]‡, **José Manuel Luque-Pinilla**[1]‡, **Daniele Santilli**[6]‡, **Domenico Sambataro**[7]‡, **Sebastiano E. Torrisi**[4]‡, **Ada Vancheri**[4]‡, **Marwin Gutiérrez**[5]‡, **Mayra Mejia**[8]‡, **Stefano Palmucci**[9]‡, **Flavio Mozzani**[6]‡, **Jorge Rojas-Serrano**[8]‡, **Carlo Vanchieri**[4]☉, **Nicola Sverzellati**[2]☉, **Alarico Ariani**[6]☉

1 Internal Medicine, Complejo hospitalario Ruber Juan Bravo, Universidad Europea (Madrid), Madrid, Spain, 2 Scienze Radiologiche, Dipartimento di Medicina e Chirurgia (DiMeC), University of Parma, Parma, Italy, 3 Internal Medicine, Hospital universitario Fuenlabrada, Fuenlabrada, Spain, 4 Regional Referral Center for Rare Lung Diseases, A. O. U. "Policlinico-Vittorio Emanuele" Dpt. of Clinical and Experimental Medicine, University of Catania, Catania, Italy, 5 Division of Musculoskeletal and Rheumatic Disorders, Instituto Nacional de Rehabilitación—"Luis Guillermo Ibarra Ibarra", Mexico City, Mexico, 6 Internal Medicine and Rheumatoloy Unit, Azienda Ospedaliero-Universitaria di Parma, Parma, Italy, 7 Department of Clinical and Experimental Medicine, Internal Medicine Unit, Cannizaro Hospital, University of Catania, Catania, Italy, 8 Interstitial Lung Disease and Rheumatology Unit, Instituto Nacional de Enfermedades Respiratorias, Ismael Cosio Villegas, Mexico City, Mexico, 9 Department of Medica Surgical Sciences and Advanced Technologies "GR Ingrassia", Radiology I unit, University of Catania, Catania, Italy

☉ These authors contributed equally to this work.
‡ These authors also contributed equally to this work.
* pablogvasco@gmail.com

**Data Availability Statement:** A public repository of the data will be held in: www.datadryad.org/stash/ (DOI https://doi.org/10.5061/dryad.sbcc2fr22). Direct download of the data are available from:

## Abstract

### Background

Interstitial lung disease (ILD) is a frequent manifestation of Sjögren's syndrome (SS), an autoimmune disease of salivary and lacrimal glands, and affects approximately 20% of patients. No clinical or serological features appear to be useful to predict its presence, severity or progression, and chest high-resolution computed tomography (CT) remains the gold standard for diagnosis. Semiquantitative CT (SQCT) based on visual assessment (Goh and Taouli scoring) can estimate ILD extent, although it is burdened by relevant intra- and inter-observer variability. Quantitative chest CT (QCT) is a promising alternative modality to assess ILD severity.

### Aim

To determine whether QCT assessment can identify extensive or limited lung disease in patients with SS and ILD.

### Methods

This multi-center, cross-sectional and retrospective study enrolled patients with SS and a chest CT scan. SQCT assessment was carried out in a blinded and centralized manner to calculate both Goh and Taouli scores. An operator-independent analysis of all CT scans

https://datadryad.org/stash/share/1HiLI_7GxvhUu2wIAQNCCUK5ScptshXoJuCggR4g9Js.
An independent researcher could contact with the Ethics Committee (E-Mail: ceic@fjd.es), for full access to confidential data, after requesting personal authorization.

**Funding:** This work was supported by Internal/own funds of the Complejo Hospitalario Ruber Juan Bravo for clinical research. The funder had no role in study design, data collection and analysis, decision to publish, or preparation of the manuscript.

**Competing interests:** Prof. Carlo Vanchieri is part of F. Hoffmann-La Roche Ltd. Scientific Board. He has received consulting fees and/or speaker fees from Astrazeneca, Boehringer Ingelheim, Chiesi, F. Hoffmann-La Roche Ltd and Menarini. Prof. Stefano Palmucci has reveived personal fees and honoraria for lectures from Boehringer Ingelheim, Delphi International srl and F. Hoffmann-La Roche Ltd. He has been included in the scientific board for Boehringer Ingelheim. This does not alter our adherence to PLOS ONE policies on sharing data and materials. None of the other authors have any potential conflicts of interest to disclose in relation to this work.

with the open-source software platform Horos was used to evaluate the QCT indices. Patients were classified according to the extent of ILD and differences in QCT index distribution were investigated with non-parametric tests.

## Results

From a total of 102 consecutive patients with SS, the prevalence of ILD was 35.3% (36/102). There was a statistically significant difference in QCT index distribution between the SS with ILD and SS without ILD groups (p<0.001). Moreover, SS-ILD patients with ILD >20% (by Goh score) had a QCT index statistically different from those with limited ILD extent (p<0.001). Finally, QCT indices showed a moderate-to-good correlation with the Goh and Taouli scores (from 0.44 to 0.65; p<0.001).

## Conclusions

QCT indices can identify patients with SS and ILD and discriminate those with lesser or greater lung disease.

## Background

Sjögren's syndrome (SS) is systemic autoimmune disease characterized by chronic lymphocytic inflammation of ductal epithelial structures. The disease affects principally the exocrine glands, in particular the lacrimal and salivary glands, causing dryness of mucosal surfaces–termed sicca syndrome–and glandular parenchymal damage. The pathogenesis of SS is poorly understood, and manifestations are heterogeneous [1]. Accordingly, there is increasing interest in its systemic involvement [2] and how to reach a precise diagnosis using molecular profiling [3–4].

Interstitial lung disease (ILD) remains one of most frequent pulmonary complications in primary SS and sub-clinical disease is even more common [5–6]. Pulmonary disease is a singular clinical and histopathological scenario among the organ-specific SS involvement and leads to increased risk of mortality [7–8]; and it is considered in the high systemic activity domain of the European League against Rheumatism (EULAR) Sjögren's syndrome disease activity index (ESSDAI).

An accurate differential diagnosis of ILD, obstructive disease, bronchial hyper-responsiveness, bronchiolitis, bronchiectasis or xerotrachea can be challenging [9]. High-resolution chest computer tomography (HRCT) is currently considered the gold standard imaging modality in confirming a diagnosis of SS and has clear improvements over classical chest x-ray [10–12]. It also allows for the discrimination and estimation of the extent of the ILD and can inform on treatment decisions. Likewise, semiquantitative CT (SQCT) assessment, such as the Goh [13] and Tauli [14] visual scores, can estimate ILD severity–in terms of limited or extensive disease–in autoimmune systemic diseases. However, these classification systems suffer from intra- and interobserver variability [15]. Quantitative CT (QCT) analysis is a promising tool to assess primary or secondary ILD and its extent. It is based on software providing highly accurate and operator-independent measurements, termed QCT indices [16], which have been rapidly developed and validated in specific clinical settings, including systemic autoimmune diseases [17].

To the best of our knowledge, there have been no studies focused on the use of QCT for SS with ILD. Herein, we designed a study to evaluate whether QCT indices have clinical utility to screen for ILD and to appraise differences between limited and extensive ILD.

## Methods

We performed a multicenter, cross-sectional, and retrospective study in patients with SS enrolled in four university-affiliated centers.

The following inclusion criteria were applied: 2016 American College of Rheumatology (ACR)/EULAR criteria [18], a chest CT scan ordered by a primary care physician for any reason, and age older than 18 years. The exclusion criteria included those specified by the 2016 ACR/EULAR consensus, including IgG4 disease, any immunosuppressive therapy with any biological agent in the last 3 months, or methotrexate or leflunomide in the year prior to study inclusion. Prednisone (or equivalent) at a dose ≤7.5 mg (on a tapering plan only) or low-dose hydroxychloroquine was allowed. The research protocol was approved by the local ethics committees and was conducted in accordance with the tenets of the Helsinki Declaration. The protocol was developed following the STROBE statement [19]

The following data were collected from all patients: demographic variables (age, sex), date of disease onset, symptoms suspicion of pulmonary disease, smoking habit, chest CT scan, pulmonary function tests [diffusing capacity of carbon monoxide (DLco), DLco divided by alveolar volume (DLco/VA), forced vital capacity (FVC) and total volume capacity (TLC)], and autoantibodies profile (SSA/Ro, SSB/La). The laboratories of all the participating centers used the same methodology, which was adopted from the current standards of the American Thoracic Society/European Respiratory Society.

A DICOM (digital imaging and communications in medicine) viewer, open-source software (Horos **www.horosproject.org**) was used for analyses and the following QCT indices were obtained after scan processing: kurtosis (Kurt), mean lung attenuation (MLA), skewness (Skew) and standard deviation (Sdev). The entire procedure including the lung segmentation algorithm was performed as described previously [16]. Accordingly, the region of interest between -950 HU and 400 HU was considered as 'pulmonary parenchyma' (namely, lung parenchyma without vessels or bronchioles and not affected by fibrosis). Those voxels included in the whole lung volume with higher HU values were identified as non-parenchymal structures. The QTC indices were calculated according to these definitions, as parenchymal (i.e., pKurt, pSkew, etc.) and total (i.e., tKurt, tSkew, etc.) QTC. These indices are based on the histograms obtained from the computed analysis of the volumetric region of interest [17, 20].

All CT images were centrally and blindly reviewed by the same board-certified radiologists (MS, NS) and QCT indices were then calculated. The SQCT assessment was carried out to calculate both Goh and Taouli scores for each CT scan. On the basis of the SQCT assessment, two groups were established: patients with SS and with/without ILD.

Data were reported as mean and SD for continuous variables and numbers and percentages for categorical variables. The Kolgomorov-Smirnov test was used to check the assumption of normality of the continuous variables. Differences between subgroups were analyzed using a non-parametric test, as appropriate. The Spearman rank test was used to determine correlations of QCT indices with SQCT scores, pulmonary function tests, and the other collected variables. A Youden index, according to the area under the curve, was run to select the best cut-off point value of QCT indices to assess the ILD.

The research was approved by the ethics committee of Fundación Jiménez Díaz, Madrid. Approval Number: PIC 17–2017 The data were analyzed anonymously.

A two-tailed probability of p<0.05 was considered statistically significant. All analyses were conducted using SPSS 20.0 (Chicago, IL, USA) and R (version 3.5.2) statistical software package (http://www.R-project.org/) (Vienna, Austria).

## Results

A total of 102 patients were enrolled between January 2017 and September 2018. Patient characteristics are listed in Table 1.

The diseases associated to the development of SS, secondary SS, were: rheumatoid arthritis (5,9%), systemic lupus erythematosus (2%), systemic sclerosis (12,7%), and undifferentiated connective tissue disease (9,8%). Of the, just 10 cases have associated ILD.

Pulmonary function tests were incomplete in more than half of all patients (41% and 58% of patients did not have FCV and DLco data, respectively). No differences were found for patients with SS with ILD (SS-ILD) and those without ILD in terms of age, disease duration and autoimmune profile. The most common onset symptom in the SS-ILD group was dyspnea (52%), whereas mouth or eye dryness was the most common onset symptom in the SS without ILD group (59%). Pulmonary function tests showed that %FCV and %DLco were lower in the SS-ILD group than in the SS without ILD group (p = 0.03 and p = 0.01, respectively). As expected, there was a strong correlation between the Goh and Taouli scores (rho = 0.98; p<0.001). Table 2.

**Table 1. Baseline characteristics of the patients with Sjögren's syndrome.**

|  | Total cohort | SS without ILD | SS-ILD | p-value |
|---|---|---|---|---|
| N | 102 | 66 | 36 | - |
| Age, median (yrs) (95% CI) | 69 (65–71) | 68 (63–71) | 69 (63–74) | nss |
| Sex (M:F) | 7:95 | 3:63 | 4:32 | nss |
| Smoker (no:former:yes)* | 74:8:11 | 47:8:4 | 27:0:7 | nss |
| Disease duration, median (yrs) (95% CI) | 4 (3–5) | 3 (2–5) | 5 (3–7) | nss |
| pSS prevalence (%) | 79 | 83 | 72 | nss |
| Onset symptoms (sicca:dyspnoea:other) | 51:27:24 | 41:7:18 | 10:20:6 | <0.001 |
| Antibodies Ro/SSA prevalence (%) | 52 | 47 | 63 | nss |
| Antibodies La/SSB prevalence (%) | 25 | 21 | 31 | nss |
| FVC (%) (95% CI) ** | 97 (93–108) | 108 (93–116) | 94 (83–99) | 0.03 |
| DLco (%) (95% CI) *** | 71 (62–81) | 82 (73–87) | 63 (53–71) | 0.01 |
| ILD pattern (NSIP:UIP:Other) | - | - | 27:6:3 | - |
| Goh score (95% CI) | 0 | 0 | 12,5 (7.7–25.3) | - |
| Taouli score (95% CI) | 0 | 0 | 8,0 (5.7–11.0) | - |

Abbreviations: M, male; F, female; CI, confidence interval; pSS, primary SS; ILD, interstitial lung disease; Nss, not statistically significant; FVC, forced vital capacity; DLCO, diffusion of lung CO; TLC, total lung capacity; FEV1, forced expiratory volume in first second; NSIP, non-specific interstitial pneumonia; UIP, usual interstitial pneumonia.

* 9/102 patients data missing.

** 41/102 patients data missing.

*** 58/102 patients data missing

All QCT indices (with the exception of tSdev) had a good correlation with the Goh and Taouli scores (rho ranges from 0.36 to 0.65; p<0.001). The QCT indices (except for tSdev) strength of correlation with FVC and DLco ranged from moderate to good (rho from 0.33 to 0.55 and from 0.39 to 0.61, respectively; p<0.001). Data are reported in Table 1.

Both Goh and Taouli scores had a moderate strength of correlation with FVC (rho = -0.36 and -0.38, respectively; p<0.004) and with DLco (rho = -0.42 and -0.44, respectively; p<0.004).

In the SS-ILD group, 44% (16/36) of patients had extensive lung disease (Goh score ≥20%). These patients had similar characteristics to those with limited SS-ILD (Table 3), except for lower FVC and DLco values, and a lower prevalence of nonspecific interstitial pneumonia (69% *vs* 80%).

All QCT indices except tSDev had a different distribution in the SS-ILD *versus* SS without ILD (p<0.001) group–defining the groups as follows: 0, SS non-affected; 1, SS limited ILD; and 2, SS extensive ILD. After clustering the SS-ILD patients according to ILD extent, the QCT indices (except for tSDev) had a statistically different distribution in the three subgroups (Fig 1 and Fig 2).

Of all QCT indices, tSkew and tKurt were the best ones to differentiate ILD pattern, or not, according to AUC, 0.87 (CI95% 0.79–0.94) and 0.84 (CI95% 0.76–0.93), respectively (Table 4).

## Discussion

To the best of our knowledge, this is the first study showing that QCT indices can characterize subjects with SS -ILD as compared to the standard visual semi-quantitative methods.

Pulmonary manifestations in SS (e.g., asthenia, cough, dyspnea) are variable in intensity and severity, and are often present before a diagnosis of SS is made. The prevalence of lung involvement in SS reported in different series ranges from 12 to 61%, which underscores the clinical necessity of a correct diagnosis [21]. Moreover, abnormalities in pulmonary parenchyma can be found in up to 50% of cases and an abnormal pulmonary function test typically reflects a restrictive (lung) rather than an obstructive (airways) pattern [9]. Whereas a reduction in DLco is generally the most common abnormality, pulmonary function tests are frequently unable to correctly describe lung involvement in SS–being more accurate in the

**Table 2. Correlations of quantitative indices and semiquantitative methods and lung function tests.**

|  | pKurt | pMLA | pSdev | pSkew | tKurt | tMLA | tSdev | tSkew |
|---|---|---|---|---|---|---|---|---|
| Goh score<br>N = 102 | -0,58 | 0.48 | 0.55 | -0,56 | -0,62 | 0.44 | 0.36 | -0,65 |
| Taouli score<br>N = 102 | -0.58 | 0.48 | 0.56 | -0.55 | -0.63 | 0.45 | 0.36 | -0.65 |
| PTF FVC (%)<br>N = 61 | 0.52 | -0.55 | -0.48 | 0.52 | 0.49 | -0.41 | -0.33<br>p = 0.03 | 0.49 |
| PRF DLco<br>N = 44 | 0.61 | -0.54 | -0.58 | 0.6 | 0.39<br>p = 0.009 | -0.51 | -0.22<br>ns | 0.46<br>p = 0.002 |
| PFT TLC<br>N = 27 | 0.77 | -0.64 | -0.71 | 0.76 | 0.63 | -0.49 | -0.34<br>ns | 0.70 |
| PFT FEV1 (%)<br>N = 61 | 0.4<br>p = 0.002 | -0.43 | -0.34<br>p = 0.007 | 0.39<br>p = 0.002 | 0.48 | -0.41<br>p = 0.001 | -0.29<br>p = 0.03 | 0.49 |

All correlations have a p-value <0.001, except where specified. Abbreviations. ILD, interstitial lung disease; p(X), pulmonary quantitative indices, t(X), total quantitative indices. Kurtosis (Kurt), Sweetness (Skew), standard deviation (Sdev) and mean lung attenuation (MLA). PFT, pulmonary function test; FVC, forced vital capacity; DLCO, diffusion of lung carbon monoxide; TLC, total lung capacity; FEV1, forced expiratory volume in first second.

**Table 3. Characteristics of SS-ILD patients with limited *versus* extensive lung disease.**

| | SS-ILD cohort | SS-ILD (limited) | SS-ILD (extensive) | p |
|---|---|---|---|---|
| N | 36 | 20 | 16 | - |
| Age, median (yrs) (95% CI) | 69 (63–74) | 68 (62–76) | 70 (61–75) | nss |
| Sex (M:F) | 4:32 | 2:18 | 2:14 | nss |
| Smoke habit (no:former:yes) * | 27:0:7 | 16:0:3 | 11:0:4 | nss |
| Disease duration, median (yrs) (95% CI) | 5 (3–7) | 6 (3–12) | 4 (1–7) | nss |
| pSS prevalence (%) | 72 | 80 | 63 | nss |
| Onset symptoms (sicca:dyspnoea:other) | 10:20:6 | 5:11:4 | 5:9:2 | nss |
| SSA prevalence (%) | 63 | 70 | 56 | nss |
| SSB prevalence (%) | 31 | 35 | 25 | nss |
| FVC (%)(95% CI) ** | 94 (83–99) | 100 (86–115) | 84 (73–97) | 0.03 |
| DLCO (%)(95% CI) *** | 63 (53–71) | 70 (61–85) | 51 (47–65) | 0.02 |
| ILD pattern (NSIP:UIP:Other) | 27:6:3 | 16:1:3 | 11:5:0 | 0.05 |
| Goh score (95% CI) | 12,5 (7.7–25.3) | 7.0 (4.0–8.0) | 28.5 (25.6–46.7) | - |
| Taouli score (95% CI) | 8,0 (5.7–11.0) | 5.0 (2.2–6.0) | 13.0 (11.0–13.7) | <0.001 |

Abbreviations: M, male; F, female; CI, confidence interval; pSS, primary SS; ILD, interstitial lung disease; Nss, not statistically significant; FVC, forced vital capacity; DLCO, diffusion of lung CO; TLC, total lung capacity; NSIP, non-specific interstitial pneumonia; UIP, usual interstitial pneumonia.

* 2/36 missing data

** 7/36 missing data

*** 13/36 missing data

advanced stage of the disease–and hence have poor sensitivity to detect subclinical pulmonary involvement. Accordingly, HRCT quantification could be useful for monitoring disease, its evolution and response to therapies. The QCT indices described here provide an operator-independent assessment of lung involvement by ILD, as compared with the Taouli and Goh scores, which are operator-dependent. Indeed, the latter score results might be ambiguous in a proportion of cases, even when combined with abnormalities in pulmonary function tests, and may be unable to correctly classify some cases in the corresponding category of severity. By contrast, QCT measurements have proven to provide highly accurate and reproductible diagnoses [15], although they require some level of training to follow a standardized imaging acquisition protocol. Moreover, fully automated QCT measurements could eliminate intra- and interobserver variability, particularly when used in diagnostic decision making.

The performance of QCT indices has been previously explored in other autoimmune diseases such as systemic sclerosis (SSc). For instance, in a recent series by Ariani et al., the authors reported a moderate-to-good agreement of all values for ILD associated with SSc, and also in cases with extensive or limited lung disease. Some pulmonary function tests also showed a relatively good correlation with QCT indices (FVC and DLco <70%) (16). Moreover, QTC indices could distinguish between high and low mortality groups [22] in those cases of SSc with ILD, in relation to 1-year mortality prediction clinical scales such as ILD-GAP score (ILD subtype, gender, age, FVC and DLco) or dBi (age, history of respiratory hospitalization, and FVC value and its change after 6 months).

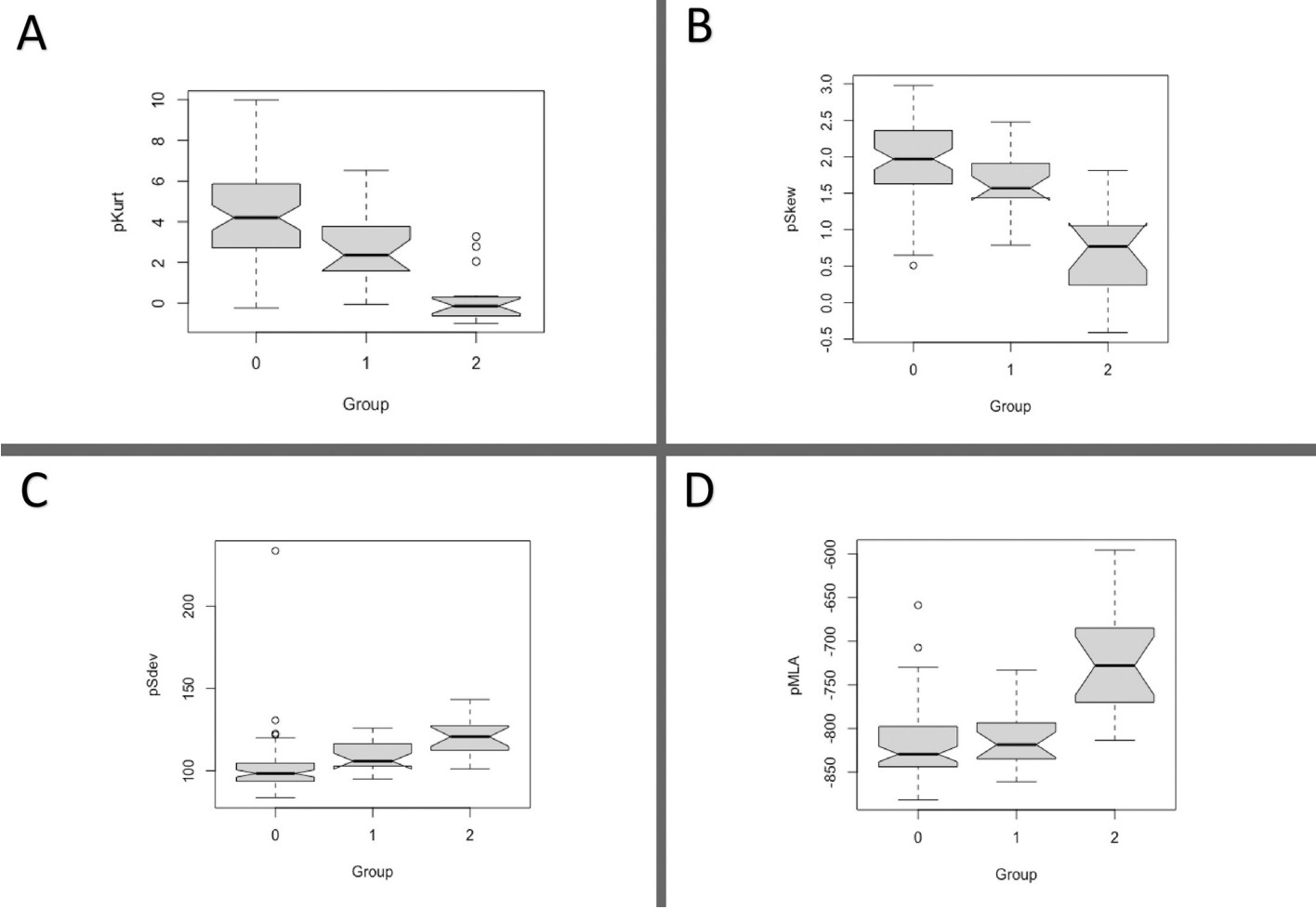

**Fig 1. Quantitative CT indices distribution in Sjögren's syndrome according to non-affected (group 0), limited ILD (group 1) and extensive (group 2) ILD.** A. Pulmonary kurtosis; B. Pulmonary skewness; C. Pulmonary standard deviation; D. Pulmonary mean lung attenuation. Differences through multiple comparisons. A. Group 0 vs 1, p = 0.011; group 1 vs 2, p = 0.003; group 0 vs 2, p< 0.001. B. Group 0 vs 1, p = 0.07; group 1 vs 2, p<0.001; group 0 vs 2, p< 0.001. C. Group 0 vs 1, p = 0.28; group 1 vs 2, p = 0.12; group 0 vs 2, p< 0.001. D. Group 0 vs 1, p = NS; group 1 vs 2, p<0.001; group 0 vs 2, p< 0.001.

Few studies have quantified pulmonary fibrosis related to SS. In the present study, we compared, for the first time, a QCT analysis of lung involvement against a specific SQCT score (Taouli) and a generic SQCT score (Goh) developed for secondary pulmonary fibrosis. The Goh score, which has been validated in SSc [13,23] and rheumatoid arthritis [24], here it shows excellent correlation with the more complex Taouli score. Hence, the Goh score appears a suitable method also for the quantification of SS-ILD. Likewise, it is reasonable to presume that the Goh score might have a similar predictive value for mortality [25, 26].

Although used as the only laboratory criteria, serologic/immunological parameters are not a definitive guide to diagnosis or to monitor the severity of ILD in SS, and the correlations across different studies are heterogenous. Indeed, we failed to find any significant association between QTC indices and anti-Ro, anti-La or ANA titers. However, anti-Ro/SSA titers were low in the general cohort but were higher in the ILD cohort.

The influence of the principal immunological markers on SS disease diagnosis was recently addressed by Brito-Zerón and colleagues using a Big Data analysis approach in 10,500 patients [27]. Regarding the phenotypes of patients, the frequency of the immunological markers ANA,

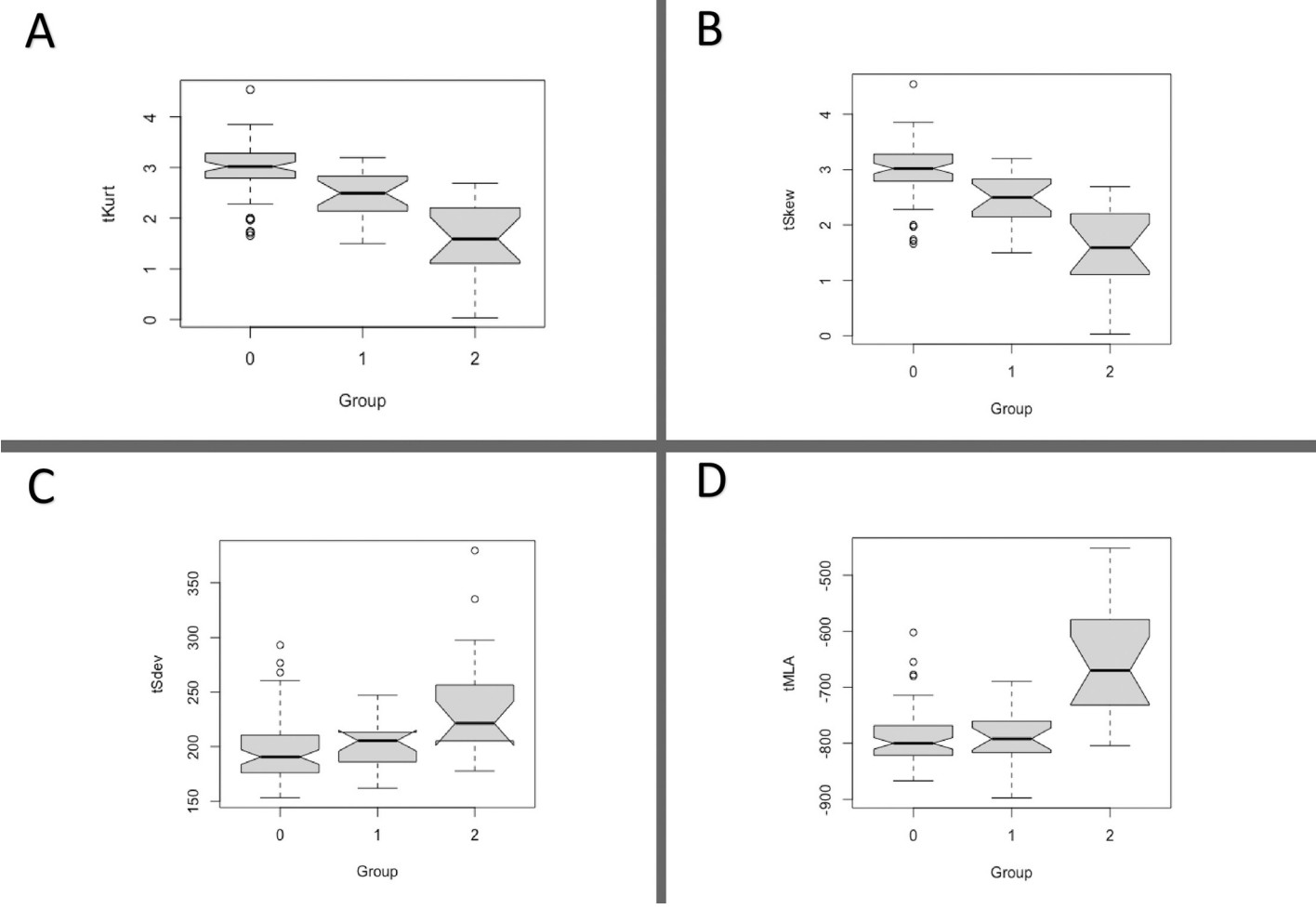

**Fig 2. Quantitative CT indices distribution in Sjögren's syndrome according to non-affected (group 0), limited ILD (group 1) and extensive (group 2) ILD.** A. Total kurtosis; B. total skewness; C. Total standard deviation; D. Total mean lung attenuation. Differences through multiple comparisons. A. Group 0 vs 1, p = NS; group 1 vs 2, p = 0.04; group 0 vs 2, p = 0.03. B. Group 0 vs 1, p = 0.001; group 1 vs 2, p<0.001; group 0 vs 2, p< 0.001.C. Group 0 vs 1, p = NS; group 1 vs 2, p = 0.004; group 0 vs 2, p< 0.001.D. Group 0 vs 1, p = NS; group 1 vs 2, p<0.001; group 0 vs 2, p< 0.001.

Ro and La were quite similar in the pulmonary domain of the ESSDAI (approximately 10%) at diagnosis. When the authors analyzed the impact of three combinations of anti-Ro/La antibodies, no statistically differences were found in the pulmonary domain. These analyses suggest that novel autoantibodies should be developed to detect ILD with sufficient sensitivity or specificity. The use of more sophisticated profiling should be incorporated as soon as possible in the daily clinical practice, for example, the ratio of blood T cells [28], or microRNA profiles [29].

In a cross-sectional study aimed to evaluate the prevalence of respiratory symptoms in SS, Kampolis et al. found that up to 20% of all cases were affected [10]. As described in the aforementioned study, it is important to differentiate those respiratory symptoms/complaints that have an onset prior to SS diagnosis, such as any underlying respiratory disease–chronic obstructive pulmonary disease, bronchial asthma, or upper chronic airway cough–as approximately one-third of these cases had an established chronic respiratory disease that preceded the onset of SS. Along this line, the impact of smoking (or former smokers) is not infrequent; however, it should be differentiated from symptoms directly related to pulmonary involvement

**Table 4. Cut-off point of quantitative indices according to the Youden index, and its corresponding sensitivity and specificity, to diagnosis interstitial lung disease in Sjögren's syndrome.**

| | Cut-off point | AUC | CI95% | Sensitivity (CI95%) | Specificity (CI95%) | Best p value |
|---|---|---|---|---|---|---|
| pKurt | 2.97 | 0.81 | 0.73–0.9 | 0.73(0.6–0.83) | 0.81(0.64–0.92) | <0.001* |
| [Δ]pMLA | -826.2 | 0.74 | 0.64–0.84 | 0.83(0.67–0.94) | 0.58(0.45–0.7) | <0.001 |
| [Δ]pSdev | 104.5 | 0.82 | 0.73–0.9 | 0.78 (0.61–0.9) | 0.71(0.59–0.82) | <0.001 |
| pSkew | 1.66 | 0.8 | 0.71–0.89 | 0.74(0.62–0.84) | 0.78(0.61–0.9) | <0.001* |
| pVol | 3318.7 | 0.67 | 0.55–0.79 | 0.71(0.59–0.82) | 0.64(0.46–0.79) | 0.004 |
| tKurt | 7.62 | 0.84 | 0.76–0.93 | 0.85(0.74–0.92) | 0.78(0.61–0.9) | <0.001* |
| tMLA[Δ] | -773.0 | 0.71 | 0.6–0.82 | 0.64(0.46–0.79) | 0.73(0.6–0.83) | <0.001 |
| tSdev[Δ] | 200.6 | 0.69 | 0.58–0.79 | 0.72(0.55–0.86) | 0.65(0.52–0.76) | 0.002 |
| tSkew | 2.72 | 0.87 | 0.79–0.94 | 0.82(0.7–0.9) | 0.81(0.64–0.92) | <0.001* |
| tVol | 2937.08 | 0.62 | 0.5–0.75 | 0.92(0.83–0.97) | 0.42(0.26–0.59) | 0.04 |

Abbreviations. ILD, interstitial lung disease; p(X), pulmonary quantitative indices, t(X), total quantitative indices. Kurtosis (Kurt), Sweetness (Skew), standard deviation (Sdev), volume (Vol) and mean lung attenuation (MLA).

* Both pKurt/pSkew and tKurt/tSkew, were statistically equivalent. [Δ] Assuming normal pulmonary patterns.

in SS. In some selected cases, however, both diseases could coexist in smokers with SS. Interestingly, the same authors did not report any specific alterations in pulmonary function tests (FEV$_1$, FVC, ratio FEV/FVC, DLco) in those patients with pulmonary disease and SS. In our series, there were some missing data for smoking habit in a small percentage of patients (<10%). Thus, we believe that the issue concerning smoking habit does not significantly impact our results.

Theoretically, pulmonary function tests might not be substantially modified in ILD in SS until the disease is more advanced. Regarding this issue, lung ultrasound showed a good performance when compared with high-resolution thorax CT [30]. Indeed, this modality could be potentially useful to detect ILD earlier in SS, independently of the patient's complaints; however, more research is needed to better understand the precise use of ultrasound as an imaging tool for identification of ILD and SS.

Our study has some limitations that should be considered, such as its retrospective design. Also, the CT protocols might be not homogenous across the participating centers–although the percentage of pulmonary disease in the global series fits well with recent data. Some differences could be expected in the immunological profile between ethnic groups and may affect the prevalence of ILD in our series [31]. Also, some degree of variable conditions when performing the CT images across the participating centers should be assumed–for example, a degree of heterogenicity in multiple slices and time points [32]. Another limitation is that while the reported pulmonary functional tests did not show obstructive profiles in most of the cases, FVC and DLco were not performed in all enrolled cases. Pulmonary function tests were often not requested at the same time as CT by the physicians. The reasons for this irregularity might be that they are not systematically assessed in SS in daily practice–considering they are likely not sufficient to help the clinician assess the extent of the ILD, or its severity. Some other biological/immunological profiles, such as cryoglobulinemia [8,27], were not collected. Our pooled analysis included primary and secondary SS, and this might impact on the QCT indices and patterns of ILD [33]. Finally, the QCT scores might be also influenced by the differences in ILD patterns in SS, such as bronchiolitis, bronchiectasis, non-specific interstitial pneumonia, usual interstitial pneumonia, lymphocytic interstitial pneumonitis, or organizing pneumonitis, among others.

QTC indices are becoming a useful tool in imaging analyses because they improve consistency of imaging diagnosis and might aid the treatment decisions in patients with ILD. This method is also promising for patient stratification according to ILD severity and extent [11, 22, 25, 26, 34]. In clinical practice, a quantitative ILD assessment with a user-friendly staging system (i.e., QCT index) could improve the outcomes of proposed SS-ILD treatments [35, 36].

Finally, the operator-independent algorithm we used in this study is free and time-saving. Accordingly, this method might be extremely suitable for multi-center trials focused on ILD. QCT indices are a promising alternative to visual scorings in ILD related to autoimmune diseases such as SS. We believe that this innovative tool will open new horizons for research into SS, as it has the capability to select ILD patients with extensive lung impairment and, correspondingly, a worse prognosis. QTC indices might potentially represent a pivotal tool at the time of diagnosis, and through management of ILD associated with SS.

## Acknowledgments

We thank K. McCreath for editorial support.

This work was included in the Departmental Project #A. Objective 2, Group of Analysis 2, "Molecular, clinical and instrumental early markers in metabolic and chronic-degenerative diseases" of the Clinical and Experimental Medicine Department, University of Catania, Catania, Italy.

## Author Contributions

**Conceptualization:** Pablo Guisado-Vasco, Mario Silva, Alarico Ariani.

**Data curation:** Pablo Guisado-Vasco, Mario Silva, Miguel Angel Duarte-Millán, Gianluca Sambataro, Chiara Bertolazzi, Mauro Pavone, Isabel Martín-Garrido, Oriol Martín-Segarra, José Manuel Luque-Pinilla, Daniele Santilli, Domenico Sambataro, Sebastiano E. Torrisi, Ada Vancheri, Marwin Gutiérrez, Mayra Mejia, Stefano Palmucci, Flavio Mozzani, Jorge Rojas-Serrano, Carlo Vanchieri, Nicola Sverzellati, Alarico Ariani.

**Formal analysis:** Pablo Guisado-Vasco, Mario Silva, Nicola Sverzellati, Alarico Ariani.

**Funding acquisition:** Pablo Guisado-Vasco, José Manuel Luque-Pinilla.

**Investigation:** Pablo Guisado-Vasco, Alarico Ariani.

**Methodology:** Pablo Guisado-Vasco, Mario Silva, Nicola Sverzellati, Alarico Ariani.

**Project administration:** Pablo Guisado-Vasco, Alarico Ariani.

**Resources:** Pablo Guisado-Vasco, José Manuel Luque-Pinilla.

**Software:** Mario Silva, Alarico Ariani.

**Supervision:** Pablo Guisado-Vasco, Alarico Ariani.

**Validation:** Pablo Guisado-Vasco, Alarico Ariani.

**Visualization:** Mario Silva, Nicola Sverzellati, Alarico Ariani.

**Writing – original draft:** Pablo Guisado-Vasco, Alarico Ariani.

**Writing – review & editing:** Pablo Guisado-Vasco, Mario Silva, Miguel Angel Duarte-Millán, Gianluca Sambataro, Chiara Bertolazzi, Mauro Pavone, Isabel Martín-Garrido, Oriol Martín-Segarra, José Manuel Luque-Pinilla, Daniele Santilli, Domenico Sambataro, Sebastiano E. Torrisi, Ada Vancheri, Marwin Gutiérrez, Mayra Mejia, Stefano Palmucci, Flavio Mozzani, Jorge Rojas-Serrano, Carlo Vanchieri, Nicola Sverzellati, Alarico Ariani.

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
