## [Decision Letter · Decision Letter 0]

21 Aug 2019

PONE-D-19-20968

Quantitative assessment of interstitial lung disease in Sjögren’s syndrome.

PLOS ONE

Dear Dr. Guisado-Vasco,

Thank you for submitting your manuscript to PLOS ONE. After careful consideration, we feel that it has merit but does not fully meet PLOS ONE’s publication criteria as it currently stands. Therefore, we invite you to submit a revised version of the manuscript that addresses the points raised during the review process.

Specifically, both reviewers found out some interests in this study, but also pointed out a number of criticisms that require improvement or even amendment. I ask the authors to fully respond to all points made by reviewers.

We would appreciate receiving your revised manuscript by Oct 05 2019 11:59PM. To enhance the reproducibility of your results, we recommend that if applicable you deposit your laboratory protocols in protocols.io, where a protocol can be assigned its own identifier (DOI) such that it can be cited independently in the future. For instructions see: http://journals.plos.org/plosone/s/submission-guidelines#loc-laboratory-protocols

We look forward to receiving your revised manuscript.

Kind regards,

Masataka Kuwana, MD, PhD

Academic Editor

PLOS ONE

Journal Requirements:

Prof. Carlo Vancheri is part of F. Hoffmann-La Roche Ltd. Scientific Board. He has received consulting fees and/or speaker fees from Astrazeneca, Boehringer Ingelheim, Chiesi, F. Hoffmann-La Roche Ltd and Menarini.

Prof. Stefano Palmucci has reveived personal fees and honoraria for lectures from Boehringer Ingelheim, Delphi International srl and F. Hoffmann-La Roche Ltd. He has been included in the scientific board for Boehringer Ingelheim.

None of the other authors have any potential conflicts of interest to disclose in relation to this work.

Reviewers' comments:

Reviewer's Responses to Questions

**Comments to the Author**

1. Is the manuscript technically sound, and do the data support the conclusions?

Reviewer #1: Partly

Reviewer #2: Yes

2. Has the statistical analysis been performed appropriately and rigorously? 

Reviewer #1: No

Reviewer #2: Yes

3. Have the authors made all data underlying the findings in their manuscript fully available?

Reviewer #1: Yes

Reviewer #2: Yes

4. Is the manuscript presented in an intelligible fashion and written in standard English?

Reviewer #1: Yes

Reviewer #2: Yes

5. Review Comments to the Author

Reviewer #1: The authors reported quantitative assessment of interstitial lung disease in Sjögren’s syndrome. It is interesting to focus the usefulness of quantitative CT in Sjögren’s syndrome-ILD, but I have some major concerns about the data

Major;

1. CTD-ILD and quantitative CT scores have already been studied, but most patients were scleroderma. Therefore, it was very interesting to examine the quantitative CT score for Sjogren's syndrome in a large number of people. However, only association between CT analysis by histogram and ILD type are not appropriate as a paper.

Regarding scleroderma, CT scores and prognosis have already been examined. Did you classify ILD as limited and extensive, but was it related to prognosis as well as scleroderma? You need to consider if CT score affects prognosis in Sjogren's syndrome.

2. The percentage of patients with pSS-ILD is 72% (Table 1).

The secondary SS needs to be described in detail.

3. It seems necessary to create a table of correlation between quantitative CT analysis and clinical findings, respiratory function tests, and semi-quantitative CT analysis.

Minor

1.The figure is difficult to see and should be considered.

2 .In figure1. 2, the statistical differences between the three groups need to be clearly stated.

Reviewer #2: The authors reported the usefulness of quantitative chest computed tomography

(QCT) assessment of interstitial lung disease (ILD) in patients with Sjögren’s syndrome (SS). In this multi-center and retrospective study, QCT indices identified patients with SS and ILD (SS-ILD), and discriminated those with lesser or greater lung disease.

This is an important study that demonstrates that QCT indices can characterize subjects with SS-ILD in comparison to the standard visual, semi-quantitative methods such as Goh and Taouli scoring.

Major

1. This study showed that QCT indices discriminated the severity of ILD in patients with SS. However, optimal cut-off points for each indicator were not determined. How QCT indices can be utilized in future research.

2. With respect to Taouli scores, in their original paper (Eur Radiol 2002; 12:1504-1511), the authors calculated scores including ground-glass attenuation, honeycombing, centrilobular nodules, reticular pattern, mosaic perfusion, and air trapping. In the present study, how were the Taouli scores determined ?

Minor

1. There were no descriptions of the correlation between QCT indices and the Goh and Taouli scores or pulmonary function test findings in Table 1. Please modify the descriptions.

6. PLOS authors have the option to publish the peer review history of their article (what does this mean?). If published, this will include your full peer review and any attached files.

Reviewer #1: No

Reviewer #2: No

---

## [Author Response · Author response to Decision Letter 0]

26 Sep 2019

Dear Editor, 

 Thank you for considering our manuscript, ‘Quantitative assessment of interstitial lung disease in Sjögren’s syndrome’ (PONE-D-19-20968), sent to PLOS One.

We really appreciate the opportunity to review our manuscript in order to fully address your concerns and comments of the referee. 

 Then, we are going to address point by point the reviewer’s recommendations. 

Journal Requirements:

I have reviewed the POLS ONE’s style guidelines, as you pointed out.

First, I have added a title page to the manuscript body, according to the format recommended by your editorial office. 

Second, I have done some changes in the body of the template/manuscript according to them -mainly headings, as required. 

Prof. Carlo Vancheri is part of F. Hoffmann-La Roche Ltd. Scientific Board. He has received consulting fees and/or speaker fees from Astrazeneca, Boehringer Ingelheim, Chiesi, F. Hoffmann-La Roche Ltd and Menarini.

Prof. Stefano Palmucci has reveived personal fees and honoraria for lectures from Boehringer Ingelheim, Delphi International srl and F. Hoffmann-La Roche Ltd. He has been included in the scientific board for Boehringer Ingelheim.

None of the other authors have any potential conflicts of interest to disclose in relation to this work.

I confirm, it does not alter the conflict of interest policy of your journal. I have added the following sentence, as recommended: ‘This does not alter our adherence to PLOS ONE policies on sharing data and materials’.

Reviewer #1: The authors reported quantitative assessment of interstitial lung disease in Sjögren’s syndrome. It is interesting to focus the usefulness of quantitative CT in Sjögren’s syndrome-ILD, but I have some major concerns about the data

Major;

1. CTD-ILD and quantitative CT scores have already been studied, but most patients were scleroderma. Therefore, it was very interesting to examine the quantitative CT score for Sjogren's syndrome in a large number of people. However, only association between CT analysis by histogram and ILD type are not appropriate as a paper.

Regarding scleroderma, CT scores and prognosis have already been examined. Did you classify ILD as limited and extensive, but was it related to prognosis as well as scleroderma? You need to consider if CT score affects prognosis in Sjogren's syndrome.

Before any new diagnostic technique can be used as an outcome measurement instrument, it should be tested its reliability comparing to actual standards. It is the aim of the present research. Later, it could be used as a prognostic tool – it is our next step in the research. In fact, the development of the quantitative indices in systemic sclerosis followed up the same idea of our working group – the count with some members whose developed this scoring system in scleroderma. First, it was release a publication assessing the reliability of these indices (2015), and them, a model of prediction of mortality was developed (2017).

References 

Jousse-Joulin S, et al. Video clip assessment of a salivary gland ultrasound scoring system in Sjögren’s syndrome using consensual definitions: an OMERACT ultrasound working group reliability exercise. Ann Rheum Dis 2019;78:967–973 

Ariani A et al. Operator-independent quantitative chest computed tomography versus standard assessment of interstitial lung disease related to systemic sclerosis: a multi-center study. Mod Rheumatol 2015; 25(5):724-30.

Ariani A et al. Quantitative chest computed tomography is associated to two prediction models of mortality in interstitial lung disease related to systemic sclerosis. Rheumatology (Oxford), 2017; 56(6): 922-927. 

2. The percentage of patients with pSS-ILD is 72% (Table 1). The secondary SS needs to be described in detail.

 A paragraph was added at the beginning of the results section, detailing the diseases associated to SS. 

The diseases associated to the development of SS, secondary SS, were: rheumatoid arthritis (5,9%), systemic lupus erythematosus (2%), systemic sclerosis (12,7%), and undifferentiated connective tissue disease (9,8%). Not all cases were affected of ILD, of course: just 10 have secondary SS.

3. It seems necessary to create a table of correlation between quantitative CT analysis and clinical findings, respiratory function tests, and semi-quantitative CT analysis.

This table has been inserted in the text – previously it was designed as supplementary material. It has been named as table 2. Hence, the former table 2 as labeled now as table 3. 

Minor

1.The figure is difficult to see and should be considered.

The figures have been edited and introduced some changes to maximize its resolution. 

2 .In figure1. 2, the statistical differences between the three groups need to be clearly stated.

 Thank you very much for this comment. In order to a better understanding and simplified of the images, it has been added a footnote in both set of figures, which explains the differences between every group (0,1,2), and corresponding p-values. 

Reviewer #2: The authors reported the usefulness of quantitative chest computed tomography

(QCT) assessment of interstitial lung disease (ILD) in patients with Sjögren’s syndrome (SS). In this multi-center and retrospective study, QCT indices identified patients with SS and ILD (SS-ILD), and discriminated those with lesser or greater lung disease.

This is an important study that demonstrates that QCT indices can characterize subjects with SS-ILD in comparison to the standard visual, semi-quantitative methods such as Goh and Taouli scoring.

Major

1. This study showed that QCT indices discriminated the severity of ILD in patients with SS. However, optimal cut-off points for each indicator were not determined. How QCT indices can be utilized in future research.

It was inserted/added a new table (number 4) to sum up these cut-off points, according to the Youden index applying using the model of area under the curve. A brief comment has been inserted in the results section. 

I am agreeing with the point, that these cut-off points could be use in the next step of the research as predictive models of mortality in ILD related to SS. 

2. With respect to Taouli scores, in their original paper (Eur Radiol 2002; 12:1504-1511), the authors calculated scores including ground-glass attenuation, honeycombing, centrilobular nodules, reticular pattern, mosaic perfusion, and air trapping. In the present study, how were the Taouli scores determined ?

These scores were calculating by two expert radiologists, properly trainee in semiquantitative scores. Hence, the score was calculated by the sum of all the categories that represent lung fibrosis, namely: honeycombing, reticular pattern, and ground-glass attenuation.

Minor

1. There were no descriptions of the correlation between QCT indices and the Goh and Taouli scores or pulmonary function test findings in Table 1. Please modify the descriptions.

 A new table (Table 2) has been inserted where the correlations are described as recommended. It is mentioned in the results paragraph. 

We hope that these comments and improvements have fulfilled the referees’ recommendations. 

Such way, your editorial office could consider the current version of the manuscript for its final acceptance.

If there are any further questions or comments after this revision, please do not hesitate to contact us. 

Sincerely, 

Pablo Guisado Vasco

---

## [Decision Letter · Decision Letter 1]

22 Oct 2019

Quantitative assessment of interstitial lung disease in Sjögren’s syndrome.

PONE-D-19-20968R1

Dear Dr. Guisado-Vasco,

We are pleased to inform you that your manuscript has been judged scientifically suitable for publication and will be formally accepted for publication once it complies with all outstanding technical requirements.

With kind regards,

Masataka Kuwana, MD, PhD

Academic Editor

PLOS ONE

Additional Editor Comments (optional):

Reviewers' comments:

Reviewer's Responses to Questions

**Comments to the Author**

1. If the authors have adequately addressed your comments raised in a previous round of review and you feel that this manuscript is now acceptable for publication, you may indicate that here to bypass the “Comments to the Author” section, enter your conflict of interest statement in the “Confidential to Editor” section, and submit your "Accept" recommendation.

Reviewer #2: All comments have been addressed

2. Is the manuscript technically sound, and do the data support the conclusions?

Reviewer #2: Yes

3. Has the statistical analysis been performed appropriately and rigorously? 

Reviewer #2: Yes

4. Have the authors made all data underlying the findings in their manuscript fully available?

Reviewer #2: Yes

5. Is the manuscript presented in an intelligible fashion and written in standard English?

Reviewer #2: Yes

6. Review Comments to the Author

Reviewer #2: Thank you for addressing my comments from initial review.The data are potentially interesting and worthy of evaluation for SS-ILD.

7. PLOS authors have the option to publish the peer review history of their article (what does this mean?). If published, this will include your full peer review and any attached files.

Reviewer #2: No

---

## [Editor Report · Acceptance letter]

1 Nov 2019

PONE-D-19-20968R1 

Quantitative assessment of interstitial lung disease in Sjögren’s syndrome 

Dear Dr. Guisado-Vasco:

I am pleased to inform you that your manuscript has been deemed suitable for publication in PLOS ONE. Congratulations! Your manuscript is now with our production department. 

With kind regards,

on behalf of

Prof. Masataka Kuwana 

Academic Editor

PLOS ONE